# Correlations between distal sensorimotor polyneuropathy and cardiovascular complications in diabetic patients in the North-Eastern region of Hungary

Attila Pető[1,2], László Imre Tóth[1], Marcell Hernyák[1], Hajnalka Lőrincz[1], Ágnes Molnár[1], Attila Csaba Nagy[3], Miklós Lukács[2], Péter Kempler[4], György Paragh[1], Mariann Harangi[1], Sztanek Ferenc[1]*

1 Department of Internal Medicine, University of Debrecen Faculty of Medicine, Debrecen, Hungary, 2 Third Department of Internal Medicine, Semmelweis Hospital of Borsod-Abauj-Zemplen County Central Hospital and University Teaching Hospital, Miskolc, Hungary, 3 Department of Health Informatics, Faculty of Health Sciences, University of Debrecen, Debrecen, Hungary, 4 Department of Internal Medicine and Oncology, Semmelweis University Faculty of Medicine, Budapest, Hungary

* sztanekf@yahoo.com

**Data Availability Statement:** All relevant data are within the manuscript and its Supporting Information files.

## Abstract

Distal sensorimotor polyneuropathy (DSPN) is the earliest detectable and the most frequent microvascular complication in diabetes mellitus. Several studies have previously demonstrated correlations between cardiovascular risk factors in diabetic patients and independent risk factors for diabetic neuropathy. Our objective was to retrospectively analyze data from diabetic patients in the North-East region of Hungary who underwent neuropathy screening at the Diabetic Neuropathy Center, University of Debrecen, between 2017 and 2021. We aimed to investigate the correlations between cardiovascular risk factors and microvascular complications among patients with DSPN. The median age of the patients was 67 years, 59,6% were female, and 91,1% had type 2 diabetes. The prevalence of DSPN among the study subjects was 71.7%. A significantly longer duration of diabetes (p<0.01) was noted in patients with DSPN. Those with DSPN demonstrated a significantly higher HbA1c level (p<0.001) and a greater frequency of insulin use (p = 0.001). We observed a significantly elevated albumin/creatinine ratio (p<0.001) and a significantly lower eGFR (p<0.001) in patients with DSPN. Diabetic retinopathy exhibited a significantly higher prevalence in patients with DSPN (p<0.001). A higher prevalence of myocardial infarction (p<0.05), ischemic heart disease (p<0.001), peripheral arterial disease (p<0.05) and a history of atherosclerosis (p<0.05) was observed in patients with DSPN. In a multivariate logistic regression analysis, the following factors were independently associated with the presence of DSPN: higher HbA1c (OR:2.58, 95% CI:1.89–3.52, p<0.001), age (OR:1.03, 95% CI:1.01–1.05, p = 0.006), albumin/creatinine ratio above 3 mg/mmol (OR:1.23, 95% CI:1.06–1.45, p = 0.008), retinopathy (OR:6.06, 95% CI:1.33–27.53, p = 0.02), and composite cardiovascular endpoint (OR:1.95, 95% CI:1.19–3.19, p = 0.008). Our study revealed that age, elevated HbA1c levels, significant albuminuria, retinopathy, and cardiovascular complications may increase

**Funding:** We are indebted to the participants of this study for their cooperation. The work is supported by the National Research, Development and Innovation Office – NKFIH, grant number: K142273. This article has been supported by the European Union, co-financed by the European Social Fund [grant no. EFOP-3.6.2-16-2017-00009 title: Establishing Thematic Scientific and Cooperation Network for Clinical Research]. This work is supported by the Hungarian Diabetes Association. The funders had no role in study design, data collection and analysis, decision to publish, or preparation of the manuscript.

**Competing interests:** The authors have declared that no competing interests exist.

the risk of DSPN. Further investigation of these associations is necessary to understand the impact of patient characteristics during the treatment of diabetic neuropathy.

## Introduction

Diabetic neuropathy is one of the most important early detectable microvascular complications in diabetes mellitus, which is associated with a significant health and economic burden [1]. The worldwide prevalence of neuropathy can be estimated at 20–50% among diabetic patients [2, 3]. In the Diabetes Control and Complications Trial, the prevalence of distal symmetric polyneuropathy (DSPN) in individuals with type 1 diabetes mellitus (T1DM) was estimated to be 20% after a 20-year duration of diabetes [4]. The incidence of DSPN is observed in 10–15% of individuals newly diagnosed with type 2 diabetes mellitus (T2DM) [5], and this proportion may escalate to nearly 50% after 10 years of diabetes [6].

According to international guidelines, screening for signs and symptoms of DSPN is recommended at the time of T2DM diagnosis and subsequently on an annual basis [7, 8]. The characteristic symptoms of DSPN are resting pain in legs, paresthesia, and sensory loss, often leading to diabetic ulceration and amputation, resulting in reduced quality of life and increased mortality [1]. The most important risk factors for diabetic neuropathy are duration of diabetes and poor glycaemic control; consequently, improving glycemic levels may prove beneficial in managing diabetic neuropathy [9]. In addition, age, gender, diabetic retinopathy, and modifiable cardiovascular risk factors, such as smoking, hypertension and hyperlipidemia, have been correlated with diabetic neuropathy in previous studies [10, 11]. Early recognition of DSPN, modification of risk factors, appropriate glycaemic control, blood pressure control, maintenance of body weight, regular physical activity, treatment of hypertension and abnormal blood lipid levels can reduce the development of complications caused by diabetes mellitus [12].

According to the international guidelines, it is recommended to assess neuropathic symptoms like paresthesia, burning sensations, and pinprick pain, along with changes in hot and cold sensation, during diabetic neuropathy screening [1, 13]. During the diagnostic process and the assessment of the severity of DSPN, symptoms and signs may be assessed through the utilization of questionnaire-based methods [14, 15]. Objective assessment of neuropathic signs (vibration perception and sensory tests) and nerve conduction studies are reliable and reproducible in determining the extent and progression of neuropathy in diabetes. The evaluation of DSPN is based on typical symptoms and neuropathy screening tests (quantitative sensory testing, vibration perception threshold) [1, 16].

Previously, studies have shown possible connections between cardiovascular risk factors in diabetic patients and independent risk factors for diabetic neuropathy [2, 10, 11]. However, these correlations have not been explored in Hungarian patients with diabetes until now. Diabetic neuropathy and the risk of cardiovascular disease are associated with an increase in all-cause mortality [17]. A retrospective cohort study in Hungary has already shown the association of DSPN with all-cause mortality [18]. While the association between cardiovascular risk factors and diabetic autonomic neuropathy is well documented [9, 10], the correlation between cardiovascular risk factors and DSPN has not been extensively studied or thoroughly investigated. The prevalence of diabetic neuropathy may vary among countries and populations, and these variations might be due to large inequalities in general health status, socioeconomic status and education and possibly due to differences in genetic susceptibility. Therefore, evaluation of data from diabetic patients in the North-East region of Hungary might explore novel

aspects of diabetic neuropathy risk prediction. In this study, our objective was to retrospectively analyze data from diabetic patients in the North-East region of Hungary who underwent neuropathy screening at the Diabetic Neuropathy Center, Department of Internal Medicine, Faculty of Medicine, University of Debrecen, between 2017 and 2021. We aimed to investigate the correlations between cardiovascular risk factors and microvascular complications in the presence of DSPN.

## Materials and methods

### Enrollment of study participants

During our investigation, we assessed a total of 1237 diabetic patients who underwent neuropathy screening at the Diabetic Neuropathy Center, Department of Internal Medicine, Faculty of Medicine, University of Debrecen, between January 1, 2017 and December 31, 2021. We excluded patients in whom other causes of polyneuropathy were identified (n = 467), such as chronic alcohol consumption, hematological disorders, chemotherapy, malignant diseases, autoimmune musculoskeletal diseases, or vitamin B12 deficiency. The inclusion criterion was the availability of hospital treatment records for the patients, so those without detailed electronic medical records were excluded (n = 102). Among the remaining patients, where confirmation of distal sensorimotor polyneuropathy was obtained, patients with incomplete data corresponding to baseline characteristics were excluded (n = 47). Study design flowchart of diabetic patients with DSPN are depicted in Fig 1. The baseline characteristics of the 621 diabetic patients included in the study were extracted from the recorded data presented in free-text format. All participants provided written informed consent. The authors had access to information that could identify individual participants during or after data collection. We had access to the data for research purposes from April 1, 2019, to December 31, 2022. The study protocol was approved by the local and regional ethical committees (protocol code: 5287-2/

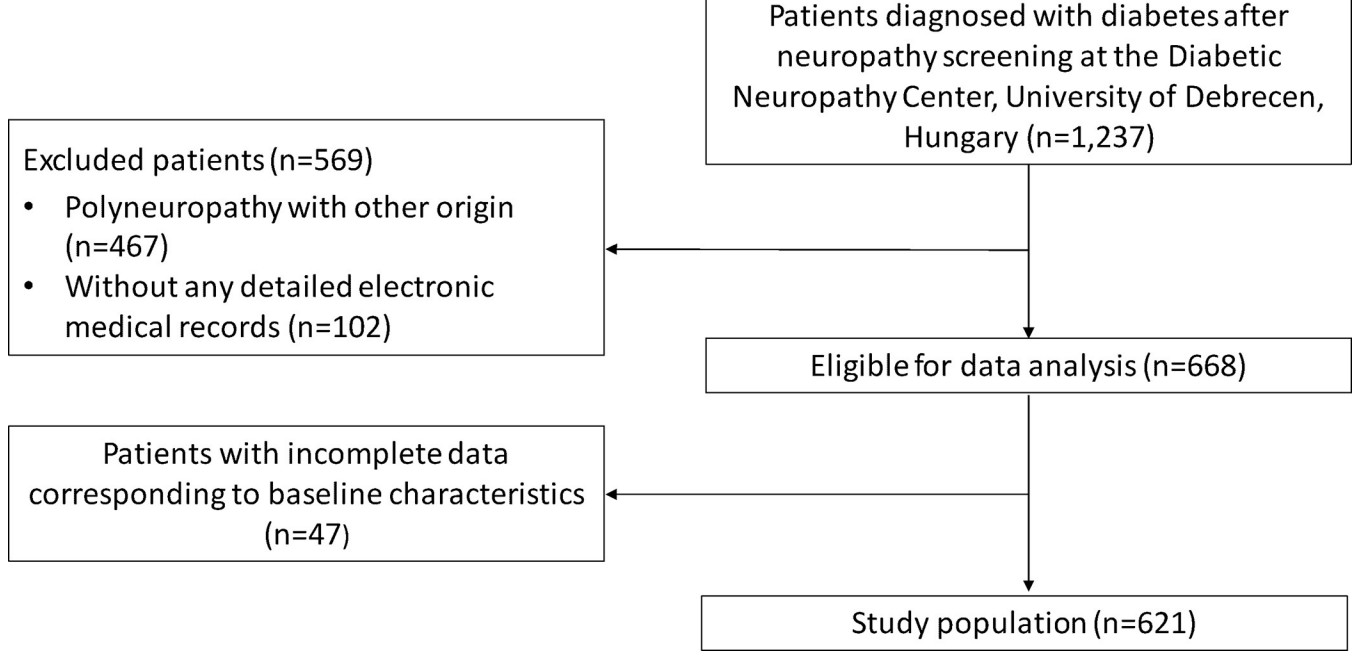

**Fig 1. Study design flowchart of diabetic patients with and without DSPN.**

2019/EKU; date of approval: 7/3/2019) and the study was carried out in accordance with the Declaration of Helsinki.

Through reviewing electronic medical records, we collected prescriptions, diagnoses, and basic medical and demographic data directly from the computerized systems of clinical and general practice. Patients with either type 1 or type 2 diabetes were identified based on the International Classification of Diseases (ICD-10) codes (E10 or E11). Our study included patients with documented diabetes duration of at least 1 year.

For each participant, we assessed age, sex, diabetes duration, type of diabetes, history of smoking and alcohol consumption, hypertension, presence of other microvascular complications (diabetic retinopathy–ICD-10: E11.5; diabetic nephropathy–ICD-10: E11.2), presence of cardiovascular diseases (coronary heart disease–ICD-10: I20, I24, I25; history of myocardial infarction–ICD-10: I21, I22, I23, I25.2; peripheral arterial disease–ICD-10: E11.5, I73.9; heart failure–ICD-10: I50; and history of stroke–ICD-1: I63, I64, G45), admission hemoglobin A1c (HbA1c), estimated glomerular filtration rate (eGFR), albumin/creatinine ratio, triglycerides, total cholesterol, low-density lipoprotein–cholesterol (LDL-C), and high-density lipoprotein–cholesterol (HDL-C). In addition, antidiabetic treatment (oral antidiabetic agents, insulin, or both) and lipid-lowering drugs were assessed. The term hepatopathy referred to non-alcoholic fatty liver disease (ICD-1: K76.0) and alcohol-induced chronic liver damage (ICD-1: K70) in the affected patients.

The diagnosis of DSPN was established according to the recommendations of the Toronto Expert Panel on Diabetic Neuropathy [7]. In our study, we selected diabetic patients in whom the typical symptoms of DSPN were observed, typically numbness, tingling, burning pain, sensitivity to touch, muscle weakness, and coordination disorders. A detailed evaluation of DSPN was performed in all participants using the DN4 (Douleur Neuropathique 4 Questions) and Neuropathy Symptom Score questionnaires during the screening of neuropathic pain. Diabetic neuropathy was defined as scoring 3 points or higher on the Neuropathy Symptom Score or 4 points or higher on the DN4 questionnaire. The superficial sensation was evaluated with a monofilament test, the vibration detection with a tuning fork test, and the examination was completed by an instrumental evaluation of the temperature detection thresholds [1, 16]. We also assessed the function of peripheral sensory nerves using a current perception threshold test with a Neurometer® (Neurotron Inc., Baltimore, Maryland, USA, 2002) [19]. The diagnosis of DSPN was established based on the presence of neuropathic symptoms or signs recorded according to neuropathy questionnaires, as well as the presence of at least one abnormal result in quantitative sensory testing, vibration perception threshold, or current perception threshold tests, in accordance with international guidelines [1, 7, 16].

## Statistical analysis

The statistical analysis was conducted using Intercooled Stata v18. (StataCorp LLC; College Station; Texas; USA). Categorical variables were presented as frequency and percentages. Pearson's chi-squared tests were employed to examine relationships between categorical variables, and, in the case of a limited number of patients, the Fisher exact test was applied. Normality was assessed using the Shapiro-Wilk test. Continuous variables were described using medians and interquartile ranges. Group comparisons were conducted using Mann-Whitney U-test or Kruskal-Wallis H test for continuous variables. In the evaluation of patients with DSPN, the occurrence of diabetic neuropathy served as the outcome variable in the statistical analysis. To assess notable differences observed among diabetic neuropathic patients, logistic regression analysis was conducted, with the presence of DSPN designated as the outcome endpoint. A multiple logistic regression model was constructed, and odds ratios with 95% confidence

intervals were reported. In a multiple logistic regression model, the selected variables included parameters for carbohydrate homeostasis (HbA1c, duration of diabetes), microvascular complications (eGFR, albumin/creatinine ratio, diabetic retinopathy), and lifestyle and macrovascular complications (composite cardiovascular endpoint, high blood pressure, smoking). The composite cardiovascular endpoint encompassed nonfatal acute myocardial infarction, chronic ischemic heart disease, nonfatal stroke, and peripheral arterial disease. In our statistical analysis, a significance level of $p < 0.05$ was considered statistically significant.

## Results

The baseline characteristics of the patients are outlined in Table 1. A total of 621 patients with diabetes mellitus participated in our retrospective study, comprising 59.6% females and 40.4% males, with a median age of 67.00 [60.00–74.00] years. Median duration of diabetes was 14.00 [6.00–27.00] years. Among the entire examined diabetic population, 86.3% had a documented history of treated hypertension, 12.6% had a history of smoking, and 57% were treated with insulin. Most of the patients received statin therapy, with a median total cholesterol level of 4.6 [3.8, 5.6] mmol/l and a median LDL-cholesterol level of 2.7 [1.9, 3.3] mmol/l. More than 90 percent of the patients were diagnosed with type 2 diabetes and the prevalence of DSPN among the studied subjects (n = 621) was 71.7%, with a count of 445 individuals affected.

The comparison of data for patients with and without neuropathy are summarized in Table 2. No differences were observed in sex, smoking status, history of hepatopathy, history of stroke, and type of diabetes between the two groups. A significantly longer duration of diabetes ($p < 0.01$) was observed in patients with neuropathy. Patients with DSPN exhibited a significantly higher HbA1c level ($p < 0.001$) and a higher frequency of insulin use (p = 0.001). We observed a significantly elevated albumin/creatinine ratio ($p < 0.001$) in patients with DSPN, while the estimated glomelural filtration rate (eGFR) was significantly lower ($p < 0.001$) compared to the group without DSPN. Higher rates of hypertension ($p < 0.01$) and dyslipidemia ($p < 0.01$) were observed in patients with DSPN. We found no significant differences in triglyceride, total cholesterol, LDL-C, and HDL-C levels between the groups with and without DSPN.

Diabetic retinopathy was significantly more prevalent in the DSPN group ($p < 0.001$). In the medical history of patients with DSPN, a higher prevalence of acute myocardial infarction ($p < 0.05$), ischemic heart disease ($p < 0.001$), heart failure ($p < 0.001$), peripheral arterial disease ($p < 0.05$), and the presence of atherosclerosis ($p < 0.05$) were demonstrated compared to individuals without DSPN.

Table 3 summarizes the results of the multivariate logistic regression analysis to explore the predictors of DSPN within our study group. Independent risk factors for DSPN included higher HbA1c (OR: 2.58, 95% CI: 1.89–3.52, $p < 0.001$); age (OR: 1.03, 95% CI: 1.01–1.05, p = 0.006); albumin/creatinine ratio above 3 mg/mmol (OR: 1.24, 95% CI: 1.06–1.45, p = 0.008); composite cardiovascular endpoint (OR: 1.95, 95% CI: 1.19–3.19, p = 0.008) and diabetic retinopathy (OR: 6.06, 95% CI: 1.33–27.53, p = 0.02). Nevertheless, no significant associations were observed between sex, eGFR, hypertension, smoking, and the risk of DSPN.

## Discussion

This is the first study that investigated the prevalence of diabetic neuropathy in the North-East region of Hungary. To our knowledge, the association between DSPN and cardiovascular composite endpoints (including nonfatal acute myocardial infarction, chronic ischemic heart disease, nonfatal stroke, and peripheral arterial disease) has not been thoroughly investigated or comprehensively studied to date. The adjusted results of the multivariate logistic regression

**Table 1. The baseline characteristics of the study population.**

| Variable | Study population N = 621 |
|---|---|
| Age (years) | 67.0 [60.0–74.0] |
| Duration of diabetes (years) | 14.0 [6.0–27.0] |
| Sex | |
| Male, n (%) | 251 (40.4) |
| Female, n (%) | 370 (59.6) |
| Type of DM | |
| T1DM, n (%) | 55 (8.7) |
| T2DM, n (%) | 566 (91.1) |
| Hepatopathy | |
| Yes, n (%) | 67 (10.8) |
| No, n (%) | 554 (89.2) |
| Hypertension | |
| Yes, n (%) | 536 (86.3) |
| No, n (%) | 85 (13.7) |
| Smoking | |
| Yes, n (%) | 78 (12.6) |
| No, n (%) | 543 (87.4) |
| Retinopathy | |
| Yes, n (%), | 39 (6.3) |
| No, n (%) | 582 (93.7) |
| Ischaemic heart disease | |
| Yes, n (%) | 138 (22.2) |
| No, n (%) | 483 (77.8) |
| History of acute myocardial infarction | |
| Yes, n (%) | 41 (6.6) |
| No, n (%) | 580 (93.4) |
| History of stroke | |
| Yes, n (%) | 50 (8.1) |
| No, n (%) | 571 (92.0) |
| History of heart failure | |
| Yes, n (%) | 90 (14.2) |
| No, n (%) | 531 (85.5) |
| History of peripheral artery disease | |
| Yes, n (%) | 53 (8.5) |
| No, n (%) | 568 (91.5) |
| History of atherosclerosis | |
| Yes, n (%) | 147 (23.7) |
| No, n (%) | 474 (76.3) |
| History of cardiovascular disease | |
| Yes, n (%) | 196 (31.6) |
| No, n (%) | 425 (68.4) |
| Insulin treatment | |
| Yes, n (%) | 354 (57.0) |
| No, n (%) | 267 (43) |
| Oral antidiabetic drugs | |
| Yes, n (%) | 568 (91.5) |
| No, n (%) | 53 (8.5) |

(*Continued*)

**Table 1.** (Continued)

| Variable | Study population N = 621 |
|---|---|
| Statin treatment | |
| Yes, n (%) | 511 (82.3) |
| No, n (%) | 110 (17.7) |
| eGFR (ml/min/1.73m$^2$) | 80.0 [72.0, 90.0] |
| HbA1c (%) | 7.3 [6.7, 8.0] |
| Triglyceride (mmol/l) | 1.6 [1.1, 2.5] |
| Total cholesterol (mmol/l) | 4.6 [3.8, 5.6] |
| HDL-C (mmol/l) | 1.3 [1.0, 1.5] |
| LDL-C (mmol/l) | 2.7 [1.9, 3.3] |

analysis indicated that age, the presense of diabetic retinopathy, microalbuminuria, higher HbA1c levels, and a composite cardiovascular endpoint were associated with an increased risk of DSPN.

In our present study, we found a significant correlation between the age of diabetic patients and the risk of DSPN. Several cross-sectional studies have supported that age is an

**Table 2. The comparison of data for diabetic patients with and without DSPN.**

| Variable | patients with DSPN | patients without DSPN | p-value |
|---|---|---|---|
| | n = 445 (71.6%) | n = 176 (28.4%) | |
| Age (years) | 68 (62.0, 74.0) | 66 (60.0, 72.0) | 0.335 |
| Duration of diabetes (years) | 14.0 (9.0, 21.0) | 12.0 (7.0, 18.0) | 0.003 |
| Sex | | | |
| Male: n (%) | 186 (41.8) | 64 (36.6) | 0.232 |
| Female: n (%) | 259 (58.2) | 111 (63.4) | |
| Type of DM (total) | | | |
| T1DM: n (%) | 39 (8.8) | 16 (9.1) | 0.888 |
| T2DM: n (%) | 405 (91.2) | 159 (90.9) | |
| eGFR (ml/min/1.73m$^2$) | 78.0 (68.0, 86.0) | 86.0 (76.0, 90.0) | <0.001 |
| HbA1c (%) | 7.5 (6.9, 8.2) | 6.7 (6.4, 7.2) | <0.001 |
| Albumin/creatinine ratio (mg/mmoll) | 3.0 (2.0, 5.0) | 2.0 (1.0, 2.0) | <0.001 |
| Triglyceride (mmol/l) Median (IQR) | 1.7 (1.1, 2.6) | 1.50 (1.1, 2.3) | 0.143 |
| Total cholesterol (mmol/l) | 4.5 (3.8, 5.6) | 4.70 (4.0, 5.4) | 0.640 |
| HDL-C (mmol/l) | 1.2 (1.0, 1.5) | 1.3 (1.0, 1.5) | 0.901 |
| LDL-C (mmol/l) | 2.7 (1.8, 3.3) | 2.6 (2.1, 3.2) | 0.550 |
| Hepatopathy, n (%) | 51 (11.5%) | 16 (9.2%) | 0.415 |
| Dyslipidaemia, n (%) | 355 (79.8%) | 119 (68.4%) | 0.003 |
| Insulin treatment, n (%) | 210 (47.2%) | 56 (32.2%) | 0.001 |
| Hypertension, n (%) | 395 (89.0%) | 138 (79.8%) | 0.003 |
| Smoking, n (%) | 58 (13.2%) | 20 (11.5%) | 0.565 |
| Retinopathy, n (%) | 37 (8.3%) | 2 (1.2%) | <0.001 |
| Ischaemic heart disease, n (%) | 118 (26.5%) | 20 (11.43%) | <0.001 |
| History of acute myocardial infarction, n (%) | 35 (7.9%) | 6 (3.4%) | 0.045 |
| History of stroke, n (%) | 36 (8.1%) | 14 (8.0%) | 0.970 |
| History of heart failure, n (%) | 79 (17.0%) | 10 (5.7%) | <0.001 |
| History of peripherial artery disease, n (%) | 45 (10.1%) | 8 (4.6%) | 0.026 |
| History of atherosclerosis, n (%) | 116 (26.1%) | 31 (17.7%) | 0.028 |

**Table 3. The results of the multivariate logistic regression analysis to explore the predictors of DSPN within our study group.**

| Factors | Odds ratio (OR) | 95% confidence interval | p-value |
|---|---|---|---|
| Sex (male/female) | 0.886 | 0.583–1.347 | 0.571 |
| Age (years) | 1.029 | 1.008–1.051 | 0.006 |
| Duration of diabetes (years) | 1.001 | 0.977–1.026 | 0.918 |
| HbA1c (%) | 2.583 | 1.893–3.524 | <0.001 |
| Albumin/creatinine ratio (mg/mmol) | 1.238 | 1.057–1.450 | 0.008 |
| eGFR (ml/min/1.73m$^2$) | 0.990 | 0.968–1.011 | 0.347 |
| Retinopathy | 6.060 | 1.334–27.528 | 0.020 |
| Composite cardiovascular endpoint | 1.949 | 1.192–3.188 | 0.008 |
| Hypertension | 0.985 | 0.540–1.798 | 0.962 |
| Smoking | 1.080 | 0.577–2.020 | 0.811 |

independent risk factor for the progression of DSPN in patients with T2DM, and this may be related to the accumulation of cellular damage over time due to increased vascular senescence and endothelial dysfunction with atherosclerosis, which may lead to axonal damage and demyelination in the progression of DSPN [20–22]. Based on a recent meta-analysis performed on a large number of patients, age was considered an independent risk factor for the development of diabetic neuropathy in T2DM [23]. This observation may be attributed to the older age and longer duration of diabetes among the neuropathic patients in the study cohort, who also exhibited a higher prevalence of other microvascular complications.

The risk of DSPN may increase with the duration of diabetes. Despite the significantly longer duration of diabetes in patients with DSPN, our multivariate logistic regression analysis did not reveal a significant association between duration of diabetes and the risk of DSPN. We hypothesize that, despite the statistical difference, both patient groups can be considered long-standing diabetics, which may explain why the duration of diabetes did not prove to be a significant predictor of DSPN in the multivariable logistic regression model. In regard to gender, there was no difference in the risk of DSPN. The present study was consistent with previous findings that reported no specific gender differences in the development of diabetic neuropathy [24–26].

In our study, we observed that patients with DSPN had significantly higher HbA1c levels compared to the non-neuropathic group. Previous studies have also confirmed the association between poor glycaemic control and the risk of diabetic neuropathy in patients with diabetes mellitus [9, 27]. Effective treatment of high serum glucose level may reduce the progression of DSPN primarily in patients with T1DM, but has limited benefits for those with T2DM [28]. The results of our study are consistent with previous findings that hyperglycemia may be significantly associated with an increased risk of diabetic neuropathy [9, 27].

We observed an association between a history of dyslipidemia and the presence of DSPN, while no correlations were found with current plasma cholesterol and triglyceride levels at the time of the DSPN diagnosis. Abnormal plasma lipid profiles are associated with an increased risk of cardiovascular disease in patients with diabetes [29]. Additionally, elevated levels of LDL-C, total cholesterol, and low levels of HDL-C were found to be significantly associated with the severity of diabetic neuropathy [30]. Elevated levels of LDL-C and serum triglyceride have also been associated with the development of DSPN in patients with T2DM [31]. The effect of lipid-lowering therapy with statins or fibrates on lipid profiles may explain why we were unable to detect any significant associations between plasma lipid profiles and DSPN in our patients with T2DM.

The development of DSPN was also associated with other potentially modifiable cardiovascular risk factors, such as hypertension and dyslipidemia, in line with previous observational

studies [2, 10, 11]. Furthermore, the association between insulin use and diabetic neuropathy is likely to be linked to the development of advanced diabetic complications, consistent with previous studies [2, 4]. It is likely that diabetes treatment is intensified more frequently among these patients to prevent further progression of complications.

The prevalence of diabetic retinopathy was significantly higher among patients with DSPN. The association between diabetic neuropathy and diabetic retinopathy can be explained by various factors. Primarily, the severity of T2DM is markedly linked to complications arising from diabetes. A prolonged duration of diabetes and the administration of insulin are indicative of more advanced cases, leading to a relatively higher prevalence of DSPN in these diabetic patients [32]. Furthermore, common pathogenic mechanisms may underlie the concurrent presence of diabetic retinopathy and nephropathy. Hyperglycemia-induced oxidative stress activates alternative cellular metabolic pathways, such as the polyol, hexosamine, and protein kinase A pathways, which are implicated in the pathogenesis of diabetic complications. Oxidative stress, resulting from the cytotoxic effects of reactive oxygen species and the suppression of the antioxidant defense system, accelerates the progression of diabetes complications such as diabetic neuropathy, retinopathy, and nephropathy [33].

Diabetic nephropathy has been identified in previous studies as a risk factor for diabetic neuropathy [32, 34]. We also found a relationship between DSPN and the presence of diabetic nephropathy and albuminuria. Additionally, patients with DSPN exhibited a lower eGFR compared to diabetic patients without neuropathy. The presence of albuminuria increased the risk of DSPN in multivariate logistic regression analysis. These findings align with studies conducted in Germany, in the UK [34], and in China as well [32]. DSPN correlated well with increased albuminuria in patients with T1DM and T2DM [35, 36]. The clinical stages of diabetic nephropathy, persistent disruption of the neurovascular unit and loss of autoregulation may all contribute to the progression of diabetic neuropathy. Consequently, changes in glomerular function in diabetes are a component of a broader spectrum of effects on the nervous system that includes sensory and autonomic neuropathies [37]. In addition, there is evidence that oxidative stress, advanced glycation end products formation, and hemodynamic changes are specifically involved in neurodegeneration during diabetic neuropathy [38].

We also identified an association between DSPN and the presence of cardiovascular disease. The composite cardiovascular endpoint, defined as non-fatal acute myocardial infarction, chronic ischemic heart disease, non-fatal stroke and peripheral arterial disease, emerged as an independent risk factor for DSPN in our study. Previous studies have confirmed the increased risk of diabetic neuropathy in individuals with diabetes who have survived myocardial infarction, additionally, DSPN has been demonstrated to correlate with an increased risk of initial cardiovascular events in T2DM [39, 40]. Cardiac autonomic neuropathy is characterized by impairment to the autonomic nerve fibers within the heart muscle, representing one of the prevalent complications in diabetes. Cardiac autonomic neuropathy results in disruptions to heart rate and vascular dynamics, thereby contributing to acute myocardial damage and the initiation of diabetic heart failure [41]. The association between DSPN and ischemic stroke is a matter of controversy. In a national cohort study, T2DM patients without microvascular complications were compared with those exhibiting diabetic retinopathy, nephropathy, or neuropathy, revealing a higher incidence of ischemic stroke. However, after adjusting for confounding factors, the differences were no longer significant [42]. An independent association between peripheral artery disease and DSPN was observed in population-based baseline surveys of the prevalence of diabetic neuropathy [43]. Based on our study, we hypothesize that early diagnosis of peripheral and autonomic neuropathy might contribute to the reduction of cardiovascular events, heart failure, and premature cardiovascular mortality in the diabetic population.

Our present study has several clinical implications. Primarily, in the diagnosis of microvascular complications in diabetes, such as retinopathy and nephropathy, there is a significant likelihood of detecting neuropathy. The association between diabetic neuropathy and cardiovascular events should be considered the most important finding of our study. Indeed, it is not surprising, as diabetic neuropathy, particularly cardiac autonomic neuropathy, is often the earliest microvascular complication to manifest in diabetes mellitus. Individuals with diabetes and cardiovascular disease should undergo screening for cardiac autonomic neuropathy and DSPN. Consenquently, the use of diabetic neuropathy screening tests may reveal peripheral nerve dysfunction in recently diagnosed diabetic patients, enabling more appropriate glycemic control and increased monitoring of other microvascular complications during diabetes care.

Strengths of the study include extensive data collection from a sufficiently large sample of DSPN patients verified through instrumental examinations, enabling a comprehensive assessment of risk factors, cardiovascular disease, and microvascular complications. However, despite these strengths, it is important to interpret this study in the context of its potential limitations. Initially, the study was designed as a retrospective single-center cohort study. Secondly, because we identified cardiovascular and microvascular complications through outpatient and inpatient hospital records, it is important to acknowledge the potential for disease underreporting, as data from primary care sources are currently not fully available. Consequently, the correlations we found were not exploratory in nature. Third, our study has limitations in regard to ethnic diversity, as it predominantly focused on the Causasian population. Therefore, interpretation of results for other ethnic groups is only possible to a limited extent. Fourth, due to gaps in the medical documentation, we were unable to explore many potential confounding factors in our study, such as lifestyle factors (diet, physical activity) and socioeconomic status. Fifth, we exclusively investigated baseline characteristics as potential risk factors for DSPN; however, some of these characteristics may change over time. The relationship between cardiac autonomic neuropathy and cardiovascular diseases is well known, but we did not have any data regarding autonomic neuropathy in the medical records. Additionally, certain variables, such as body mass index, were not included in the study but could also act as risk factors for DSPN. Sixth, as part of the diagnosis of neuropathy, we did not perform an electrophysiological examination (electroneurography) or determine the intraepidermal nerve fiber density from a skin biopsy. Finally, due to the retrospective nature of our study, any causal relationships should be approached with caution.

## Conclusion

This study represents the initial investigation into the prevalence of DSPN within the North-East region of Hungary. This study is the first to establish an association between DSPN and cardiovascular composite endpoints in patients with diabetic neuropathy. Our study verified that age, elevated HbA1c levels, the presence of significant albuminuria, retinopathy, and cardiovascular complications may increase the risk of DSPN in patients with diabetes. Given the retrospective nature of our analysis, it is imperative to conduct additional prospective studies to validate these findings. Further scrutiny of these associations is necessary to explore the influence of patient characteristics during the treatment of diabetic neuropathy.

## Supporting information

**S1 File. Minimal data set: All relevant data are within the manuscript and its supporting information file.**
(XLSX)

## Author Contributions

**Conceptualization:** Sztanek Ferenc.

**Data curation:** Attila Pető, Marcell Hernyák, Ágnes Molnár, Sztanek Ferenc.

**Formal analysis:** Attila Pető, Sztanek Ferenc.

**Investigation:** Attila Pető, László Imre Tóth, Marcell Hernyák, Hajnalka Lőrincz, Attila Csaba Nagy.

**Methodology:** Attila Pető, Attila Csaba Nagy, Sztanek Ferenc.

**Project administration:** László Imre Tóth, Ágnes Molnár, Sztanek Ferenc.

**Resources:** György Paragh.

**Software:** Attila Pető.

**Supervision:** Hajnalka Lőrincz, Miklós Lukács, Péter Kempler, Mariann Harangi, Sztanek Ferenc.

**Validation:** Attila Csaba Nagy, Sztanek Ferenc.

**Visualization:** Attila Pető, Hajnalka Lőrincz, Sztanek Ferenc.

**Writing – original draft:** Attila Pető.

**Writing – review & editing:** Sztanek Ferenc.

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
