## [Decision Letter · Decision Letter 0]

16 Apr 2024

PONE-D-24-06839Correlations between distal sensorimotor polyneuropathy and cardiovascular complications in diabetic patients in the North-Eastern region of HungaryPLOS ONE

Dear Dr. Ferenc,

Thank you for submitting your manuscript to PLOS ONE. After careful consideration, we feel that it has merit but does not fully meet PLOS ONE’s publication criteria as it currently stands. Therefore, we invite you to submit a revised version of the manuscript that addresses the points raised during the review process.

**ACADEMIC EDITOR: **

While the manuscript exhibits intriguing potential, it requires significant revisions and further refinement.<o:p></o:p>

Although the inherent interest in the subject matter is acknowledged, the reviewers have raised crucial concerns that must be properly addressed.

We look forward to receiving your revised manuscript.

Kind regards,

Marcelo Arruda Nakazone, M.D., Ph.D.

Academic Editor

PLOS ONE

Journal Requirements:

"We are indebted to the participants of this study for their cooperation. The work is supported by the National Research, Development and Innovation Office – NKFIH, grant number: K142273. This article has been supported by the European Union, co-financed by the European Social Fund [grant no. EFOP-3.6.2-16-2017-00009 title: Establishing Thematic Scientific and Cooperation Network for Clinical Research]. This work is supported by the Hungarian Diabetes Association. "

Reviewers' comments:

Reviewer's Responses to Questions

**Comments to the Author**

1. Is the manuscript technically sound, and do the data support the conclusions?

Reviewer #1: Yes

Reviewer #2: Yes

Reviewer #3: No

2. Has the statistical analysis been performed appropriately and rigorously? 

Reviewer #1: No

Reviewer #2: Yes

Reviewer #3: No

3. Have the authors made all data underlying the findings in their manuscript fully available?

Reviewer #1: Yes

Reviewer #2: Yes

Reviewer #3: No

4. Is the manuscript presented in an intelligible fashion and written in standard English?

Reviewer #1: Yes

Reviewer #2: Yes

Reviewer #3: Yes

5. Review Comments to the Author

Reviewer #1: The use of abbreviations has not been consistent throughout the text, either repeatedly spelled out again or not defined when it was first mentioned, e.g. DSPN, DN4. The authors ought to thoroughly cross check to keep the consistency.

The whole third paragraph of Introduction is redundant.

The definition of diabetic neuropathy has not been described in details in Methods. Was the NCS done for each patient? or just based on the screening questionnaires? The scoring of the questionnaires also has to be defined for clearer justification.

Suggest to reduce the characteristic values to one decimal point, as the data presented is just too messy.

What is hepatopathy? Please define. Or use a more conventional terminology for the meant character.

Repeated typo of "acut" for "acute" in the tables.

Please remove the statistical test column in Table 2 as this has been defined in your Methods (Statistical Analysis).

The data presented in Table 2 is just too messy - you can remove the "No" for each characteristics since this is not your concern.

My major concern of the paper is how come the duration of the diabetes in not included in the multivariate logistic regression model? How did the authors include the variables into the model? This has not been clearly defined in the Methods as well.

What is KORA? Again, please use and define your abbreviation appropriately.

The discussion of 2-3 pages is just too long. I would suggest the authors to trim down the discussion.

Reviewer #2: Thank you for the opportunity to review this manuscript.

The study titled "Correlations between distal sensorimotor polyneuropathy and cardiovascular complications in diabetic patients in the North-Eastern region of Hungary" investigates the relationship between distal sensorimotor polyneuropathy (DSPN) and cardiovascular complications among diabetic patients. It's a retrospective analysis of data from diabetic patients who underwent neuropathy screening at the Diabetic Neuropathy Center, University of Debrecen, between 2017 and 2021. The findings highlight a significant correlation between DSPN and various cardiovascular risk factors, including elevated HbA1c levels, albuminuria, retinopathy, and a history of cardiovascular diseases. The study emphasizes the importance of monitoring these factors in diabetic patients to mitigate the risk of developing DSPN. The authors describe a clinically relevant topic; however the study is region-specific, which might limit the generalizability of the findings to other populations with different ethnic backgrounds or healthcare systems.

I have some minor comments:

1. General comments

The retrospective design could introduce recall bias or inaccuracies in medical records, potentially affecting the results' reliability.

Due to its cross-sectional design, the study cannot identify risk factors for the development or progression of DSPN, only associations between DSPN and other vascular complications.

While the study attempts to control for various factors, other potential confounders, such as lifestyle factors (diet, physical activity) and socioeconomic status, are not explicitly addressed.

Abbreviations: please use abbreviations consistently (DSPN or DN).

2. Methods and results

Patients’ selection: how did you exclude patients with heart failure other causes, like cardiomyopathies, atrial fibrillation or structural abnormalities?

You mentioned that confirmed DSPN (based on Toronto criteria) was a inclusion criteria, but you have a group without DSPN too. How did you define neuropathic signs? Could you provide nerve conduction data? You have used Neurometer to diagnose DSPN, but you did not show any data. You assessed neuropathic pain but did not provide any data about it.

In the table of baseline characteristic, you show hepatopathy. How did you define hepatopathy? Does it mean NAFLD or NASH?

The association between DSPN and CVD is a clinically important result. However, regardless of the results obtained from the regression model, it is important to emphasize that accurate predictions cannot be guaranteed by cross sectional study. Please mention it in a limitation section.

It's a cross-sectional analysis, so you only have associations with the prevalence of DSPN but no data on the risk of DSPN.

3. Discussion

There are only few studies showing association between peripheral neuropathy and cardiovascular risk. However, a study with 30-month follow-up period could show that PN is associated with increased risk for a first cardiovascular event among individuals with type 2 diabetes. Neuropathy was nevertheless only diagnosed with a neurofilament test. Heart. 2014 Dec;100(23):1837-43.

Line 210: Effective management of diabetes can decelerate the progression of DSPN in individuals with Type 1 Diabetes (T1D), yet it only yields limited benefits for those with Type 2 Diabetes (T2D) (Cochrane Database Syst Rev. 2012;6(6):CD007543.).

Line 257: the connection btw CAN and CVD is well known, but you did not provide any data on autonomic neuropathy. It is also a limitation of the study.

Reviewer #3: Thank you for the opportunity to review this paper on the “Correlation between distal sensorimotor polyneuropathy and cardiovascular complications in diabetic patients in the North-Eastern region of Hungary. “

I enjoyed reading the paper and I think the data will be of interest to doctors locally and it serves as a good basis to build a research program.

Unfortunately it has significant limitations:

-it is a retrospective study

-high number of exclusions, criteria for exclusion are not clearly stated i.e. what is the difference between “without any detailed electronic medical records” and “incomplete electronic health documentation”.

-how was neuropathy of non-diabetic origin defined. Diabetes at diagnosis is not the same as the time living with diabetes as it is often asymptomatic.

-the criteria for inclusion are not clearly stated (apart from the reference to the Toronto criteria).

-the data for the detailed clinicals assessments are not shown including the neurological clinical examination

-current perception threshold testing using the Neurometer is not an accepted measure of neuropathy.

-no mention of a standardised and validated measures of neuropathy such as nerve conduction studies or skin biopsy for intra-epidermal nerve fibre density.

-multiple logistic regression model lacks detail on model fit statistics, the estimated coefficients for each independent variable included in the model, confounding variables and mention of assumptions.

6. PLOS authors have the option to publish the peer review history of their article (what does this mean?). If published, this will include your full peer review and any attached files.

Reviewer #1: No

Reviewer #2: No

Reviewer #3: No

---

## [Author Response · Author response to Decision Letter 0]

9 May 2024

Responses to Reviewer #1: 

Thank you for the thoughtful review and the positive comments on our manuscript. 

Responses to comments: 

- The use of abbreviations has not been consistent throughout the text, either repeatedly spelled out again or not defined when it was first mentioned, e.g. DSPN, DN4. The authors ought to thoroughly cross check to keep the consistency.

Response: The Reviewer is absolutely correct, there were inconsistencies in the use of abbreviations in the manuscript. We have addressed this by making corrections and indicating the changes in red within the text. Unfortunately, the literature is also inconsistent in the usage of the terms "diabetic neuropathy" and "distal sensorimotor polyneuropathy" in diabetes. The most common cause of diabetic neuropathy is distal sensorimotor polyneuropathy (DSPN), however, diabetic autonomic neuropathy refers to damage to the autonomic nervous system. In clinical practice, DSPN is most commonly examined, and these data are often used when making statements regarding diabetic neuropathy. In our study, we examined the relationships of the DSPN, therefore, we used the abbreviation "DSPN" in the corrections and corrected the cases where we presented our own data or when distal sensorimotor neuropathy is mentioned in the referenced literature. When we talked about diabetic neuropathy in general, or when the literature referred to diabetic neuropathy, we used the terminology “diabetic neuropathy”.

- The whole third paragraph of Introduction is redundant.

Response: We corrected redundant information in the third paragraph of the „Introduction” section (line 62-65, marked in blue), resulting in the following sentences being included: „According to the international guidelines, it is recommended to assess neuropathic symptoms like paresthesia, burning sensations, and pinprick pain, along with changes in hot and cold sensation, during diabetic neuropathy screening. During the diagnostic process and the assessment of the severity of DSPN, symptoms and signs may be assessed through the utilization of questionnaire-based methods.” In the latter part of the paragraph, we had to incorporate sentences requested by Reviewer 3 (line 67-73, marked in blue), thus we were unable to significantly reduce this section of the introduction.

- The definition of diabetic neuropathy has not been described in details in Methods. Was the NCS done for each patient? or just based on the screening questionnaires? The scoring of the questionnaires also has to be defined for clearer justification.

Response: We added the definition of DSPN to the „Methods” section: „DSPN is characterized as an asymmetrical, length-dependent sensorimotor polyneuropathy attributable to metabolic and microvascular changes consequent to prolonged exposure to chronic hyperglycemia, compounded by cardiovascular risk factors” (line 122-124, marked in blue). 

We also added these sentences to the „Methods” section: „A detailed assessment of DSPN was performed in all participants DN4 (Douleur Neuropathique 4 Questions) and Neuropathy Symptom Score questionnaires in screening for neuropathic pain syndrome, quantitative sensory testing, vibration perception threshold for the diagnosis of DSPN. Diabetic neuropathy was defined as scoring 3 points or higher on the Neuropathy Symptom Score or 4 points or higher on the DN4 questionnaire. Evaluation of superficial sensation is done with a monofilament test, vibration sensation with a tuning fork test, and instrumental assessment of vibration and temperature sensation thresholds may complement the examination (line 124-130, marked in blue).” 

Objective assessment of neuropathic signs (vibration perception and sensory tests) and nerve conduction studies are reliable and reproducible in determining the extent and progression of neuropathy in diabetes. The evaluation of DSPN is based on typical symptoms and neuropathy screening tests. Therefore, in accordance with international recommendations, additional confirmatory electrophysiological testing of nerve conduction velocity (electroneurography) is typically unnecessary in diabetic patients. Electrophysiological tests or determination of intra-epidermal nerve fibre density during skin biopsy are only justified if other etiological factors arise or if the diagnosis is unclear due to atypical neurological symptoms (Davies 2006, Ziegler 2022). We added these sentences to the „Introduction” section (line 64-73, marked in blue)

References:

Tesfaye S, Boulton AJ, Dyck PJ, Freeman R, Horowitz M, Kempler P, et al. Diabetic neuropathies: update on definitions, diagnostic criteria, estimation of severity, and treatments. Diabetes Care. 2010;33(10):2285-93. doi: 10.2337/dc10-1303. PubMed PMID: 20876709; PubMed Central PMCID: PMCPMC2945176.

Ziegler D, Tesfaye S, Spallone V, Gurieva I, Al Kaabi J, Mankovsky B, et al. Screening, diagnosis and management of diabetic sensorimotor polyneuropathy in clinical practice: International expert consensus recommendations. Diabetes Res Clin Pract. 2022;186:109063. Epub 20210920. doi: 10.1016/j.diabres.2021.109063. PubMed PMID: 34547367.

Davies M, Brophy S, Williams R, Taylor A. The prevalence, severity, and impact of painful diabetic peripheral neuropathy in type 2 diabetes. Diabetes Care. 2006;29(7):1518-22. doi: 10.2337/dc05-2228. PubMed PMID: 16801572.

- Suggest to reduce the characteristic values to one decimal point, as the data presented is just too messy.

Response: We have made the requested changes (Table 1 and Table 2, marked in red).

- What is hepatopathy? Please define. Or use a more conventional terminology for the meant character.

Response: During the review of the documentation of our patients, we selected the data into a neuropathy register, in which, in view of the lack of clear documentation regarding the causes of liver disease, we formulated the term "hepatopathy" as a summary. 67 patients (10.8%) from the examined patient group suffered from liver disease, and during the reevaluation of the data, the causes of "hepatopathy" were identified as nonalcoholic fatty liver disease (NAFLD) and alcohol-induced liver damage. Accordingly, we added the meaning of hepatopathy in the „Method” section (line 119-120, marked in blue).

- Repeated typo of "acut" for "acute" in the tables.

Response: We have corrected the typos (Table 1 and Table 2, marked in red).

- Please remove the statistical test column in Table 2 as this has been defined in your Methods (Statistical Analysis).

Response: We have made the requested changes at the reviewer's request (Table 2).

- The data presented in Table 2 is just too messy - you can remove the "No" for each characteristics since this is not your concern.

Response: We have made the requested changes at the reviewer's request (Table 2).

- My major concern of the paper is how come the duration of the diabetes in not included in the multivariate logistic regression model? How did the authors include the variables into the model? This has not been clearly defined in the Methods as well.

Response: We find the Reviewer's comment entirely justified. Based on the reviewer's concern, we recalculated our multivariate logistic regression model to include duration of diabetes and re-edited "Table 3" accordingly. Although the duration of diabetes was significantly higher in patients with DSPN, we did not find an association between the duration of diabetes and the risk of DSPN in multivariate logistic regression analysis. We added this sentence to the „Discussion” section: „The risk of DSPN may increase with the duration of diabetes. Despite the significantly longer duration of diabetes in patients with DSPN, our multivariate logistic regression analysis did not reveal a significant association between duration of diabetes and the risk of DSPN.” (line 204-208, marked in blue)

- What is KORA? Again, please use and define your abbreviation appropriately.

Response: We have corrected the sentence in „Discussion” section (line 260-262, marked in blue)

- The discussion of 2-3 pages is just too long. I would suggest the authors to trim down the discussion.

Response: In accordance with the Reviewer's opinion, we significantly reduced the "Discussion" section, marking the deleted sentences within the manuscript. Accordingly, the literature references were reduced by 4 references in line with the deleted sentences, which was also indicated in the „References” section. However, we needed to consider the feedback from Reviewer 2 and Reviewer 3 regarding the inclusion of the Discussion and Limitations sections.

Thank you again for your thorough review of our manuscript and your valuable feedback!

Responses to Reviewer #2: 

We would like to thank your valuable comments that improve our manuscript.

Responses to comments: 

- The retrospective design could introduce recall bias or inaccuracies in medical records, potentially affecting the results' reliability. Due to its cross-sectional design, the study cannot identify risk factors for the development or progression of DSPN, only associations between DSPN and other vascular complications.

Response: The Reviewer is absolutely correct. In the retrospective design, we took into account recall bias and imprecision in the medical records, which could have potentially affected the reliability of our results. We added these sentences to the „Limitation” section: „because we identified cardiovascular and microvascular complications through outpatient and inpatient hospital records, it is important to acknowledge the potential for disease underreporting, as data from primary care sources are currently not fully available. Consequently, the correlations we found were not exploratory in nature.” (line 288-291, marked in blue), and „due to the retrospective nature of our study, any causal relationships should be approached with caution.” (line 299-300, marked in blue) 

- While the study attempts to control for various factors, other potential confounders, such as lifestyle factors (diet, physical activity) and socioeconomic status, are not explicitly addressed.

Response: We find the Reviewer's comment entirely justified. We added this sentence to the „Discussion” section: „Due to gaps in the medical documentation, we were unable to explore many potential confounding factors in our study, such as lifestyle factors (diet, physical activity) and socioeconomic status.” (line 293-294, marked in blue)

- Abbreviations: please use abbreviations consistently (DSPN or DN).

Response: The Reviewer is absolutely correct, there were inconsistencies in the use of abbreviations in the manuscript. We have addressed this by making corrections and indicating the changes in red within the text. Unfortunately, the literature is also inconsistent in the usage of the terms "diabetic neuropathy" and "distal sensorimotor polyneuropathy" in diabetes. The most common cause of diabetic neuropathy is distal sensorimotor polyneuropathy (DSPN), however, diabetic autonomic neuropathy refers to damage to the autonomic nervous system. In clinical practice, DSPN is most commonly examined, and these data are often used when making statements regarding diabetic neuropathy. In our study, we examined the relationships of the DSPN, therefore, we used the abbreviation "DSPN" in the corrections and corrected the cases where we presented our own data or when distal sensorimotor neuropathy is mentioned in the referenced literature. When we talked about diabetic neuropathy in general, or when the literature referred to diabetic neuropathy, we used the terminology “diabetic neuropathy”.

- Patients’ selection: how did you exclude patients with heart failure other causes, like cardiomyopathies, atrial fibrillation or structural abnormalities?

Response: In the case of the patients selected for the examination, we were able to assess the comorbidities based on the available medical documentation, but we did not have the opportunity to organize detailed cardiology examinations. If we had cardiological findings, such as ECG, echocardiography, coronary CT results, etc., and etiological analysis of heart failure were also performed. However, reliable data were often insufficient for this purpose. Therefore, we did not investigate the etiology of heart failure, and the correlations we found were not of an exploratory nature, which we acknowledged in the "Limitation" section (line 288-291, marked in blue)

- You mentioned that confirmed DSPN (based on Toronto criteria) was a inclusion criteria, but you have a group without DSPN too. How did you define neuropathic signs? Could you provide nerve conduction data? You have used Neurometer to diagnose DSPN, but you did not show any data. You assessed neuropathic pain but did not provide any data about it.

Response: We added the definition of DSPN to the „Methods” section: „DSPN is characterized as an asymmetrical, length-dependent sensorimotor polyneuropathy attributable to metabolic and microvascular changes consequent to prolonged exposure to chronic hyperglycemia, compounded by cardiovascular risk factors (line 122-124, marked in blue). A detailed assessment of DSPN was performed in all participants DN4 (Douleur Neuropathique 4 Questions) and Neuropathy Symptom Score questionnaires in screening for neuropathic pain syndrome, quantitative sensory testing, vibration perception threshold for the diagnosis of DSPN. Diabetic neuropathy was defined as scoring 3 points or higher on the Neuropathy Symptom Score or 4 points or higher on the DN4 questionnaire. Evaluation of superficial sensation is done with a monofilament test, vibration sensation with a tuning fork test, and instrumental assessment of vibration and temperature sensation thresholds may complement the examination. The current perception threshold values measured with the Neurometer, as well as the DN4 and Neuropathy Symptom Score values, were utilized solely for establishing the diagnosis of neuropathy. However, we did not use the exact data in the present study, as we did not follow up the patients. „Objective assessment of neuropathic signs (vibration perception and sensory tests) and nerve conduction studies are reliable and reproducible in determining the extent and progression of neuropathy in diabetes. The evaluation of DSPN is based on typical symptoms and neuropathy screening tests. Therefore, in accordance with international recommendations, additional confirmatory electrophysiological testing of nerve conduction velocity (electroneurography) is typically unnecessary in diabetic patients. Electrophysiological tests or determination of intra-epidermal nerve fibre density during skin biopsy are only justified if other etiological factors arise or if the diagnosis is unclear due to atypical neurological symptoms.” We added these sentences to the „Introduction” section (line 65-73, marked in blue)

References:

Tesfaye S, Boulton AJ, Dyck PJ, Freeman R, Horowitz M, Kempler P, et al. Diabetic neuropathies: update on definitions, diagnostic criteria, estimation of severity, and treatments. Diabetes Care. 2010;33(10):2285-93. doi: 10.2337/dc10-1303. PubMed PMID: 20876709; PubMed Central PMCID: PMCPMC2945176.

Ziegler D, Tesfaye S, Spallone V, Gurieva I, Al Kaabi J, Mankovsky B, et al. Screening, diagnosis and management of diabetic sensorimotor polyneuropathy in clinical practice: International expert consensus recommendations. Diabetes Res Clin Pract. 2022;186:109063. Epub 20210920. doi: 10.1016/j.diabres.2021.109063. PubMed PMID: 34547367.

Davies M, Brophy S, Williams R, Taylor A. The prevalence, severity, and impact of painful diabetic peripheral neuropathy in type 2 diabetes. Diabetes Care. 2006;29(7):1518-22. doi: 10.2337/dc05-2228. PubMed PMID: 16801572.

- In the table of baseline characteristic, you show hepatopathy. How did you define hepatopathy? Does it mean NAFLD or NASH?

Response: During the review of the documentation of our patients, we selected the data into a neuropathy register, in which, in view of the lack of clear documentation regarding the causes of liver disease, we formulated the term "hepatopathy" as a summary. 67 patients (10.8%) from the examined patient group suffered from liver disease, and during the reevaluation of the data, the causes of "hepatopathy" were identified as non

---

## [Decision Letter · Decision Letter 1]

29 May 2024

PONE-D-24-06839R1Correlations between distal sensorimotor polyneuropathy and cardiovascular complications in diabetic patients in the North-Eastern region of HungaryPLOS ONE

Dear Dr. Ferenc,

Thank you for submitting your manuscript to PLOS ONE. After careful consideration, we feel that it has merit but does not fully meet PLOS ONE’s publication criteria as it currently stands. Therefore, we invite you to submit a revised version of the manuscript that addresses the points raised during the review process.

**ACADEMIC EDITOR: **The manuscript is interesting but will require further reworking and a major revision. While they recognize the potential interest of the subject studied, one of the reviewers raised several important issues that need to be properly addressed.

We look forward to receiving your revised manuscript.

Kind regards,

Marcelo Arruda Nakazone, M.D., Ph.D.

Academic Editor

PLOS ONE

Reviewers' comments:

Reviewer's Responses to Questions

**Comments to the Author**

1. If the authors have adequately addressed your comments raised in a previous round of review and you feel that this manuscript is now acceptable for publication, you may indicate that here to bypass the “Comments to the Author” section, enter your conflict of interest statement in the “Confidential to Editor” section, and submit your "Accept" recommendation.

Reviewer #1: (No Response)

Reviewer #2: All comments have been addressed

2. Is the manuscript technically sound, and do the data support the conclusions?

Reviewer #1: (No Response)

Reviewer #2: Yes

3. Has the statistical analysis been performed appropriately and rigorously? 

Reviewer #1: (No Response)

Reviewer #2: Yes

4. Have the authors made all data underlying the findings in their manuscript fully available?

Reviewer #1: (No Response)

Reviewer #2: Yes

5. Is the manuscript presented in an intelligible fashion and written in standard English?

Reviewer #1: (No Response)

Reviewer #2: Yes

6. Review Comments to the Author

Reviewer #1: The manuscript has improved after the correction, but I still have several comments to make.

Page 3 Line 62: You mean reduce the complications caused by diabetes or diabetic neuropathy? Please clarify.

Page 3 Line 70-74: Inappropriately elaborated here. This should be in your limitation to justify since you did not perform the tests.

The authors do not seem to understand the comments from multiple reviewers when we questioned how did they define their cohort of diabetic neuropathy. Page 5 Line 123-125 is redundant.

Page 5 Line 125-128: Grammarly incorrect. Please rephrase the whole sentence.

Page 5 Line 133-134: This is redundant since you have justified using other tests plus Neurometer to define the neuropathy.

Significant discrepancies in the values presented in text (including abstract) and tables. Please align all the values, including decimal points.

Page 8 Line 177 and Abstract: HbA1c above 7% - Is this variable continuous or categorical when included in the model? If this is categorical, this has not been clearly defined in Methods.

How did the authors decide which variables to be included in the multivariate logistic regression model? This is not clear from the statistical analysis. And, could that be the reason why they did not find the duration of diabetes to be significant? The significance of independent predictors is very much depended on the variables that you include in the model.

Page 9 Paragraph 3: In the end if the authors still did not find the duration of diabetes to be significant in multivariate analysis, please explain. This was not explained in the discussion.

Page 9 Paragraph 4: But your cohort significantly contains more T2DM than T1DM, how do you explain this?

Page 11 Line 266-267: How would it be your study implications when you did not actually perform the tests?

Reviewer #2: (No Response)

7. PLOS authors have the option to publish the peer review history of their article (what does this mean?). If published, this will include your full peer review and any attached files.

Reviewer #1: No

Reviewer #2: **Yes: **Zoltan Kender MD, PhD

---

## [Author Response · Author response to Decision Letter 1]

5 Jun 2024

Marcelo Arruda Nakazone, M.D., Ph.D.

Academic Editor

PLOS ONE

Dear Editor,

We have received the editor's response as well as the reviewer's comments and questions regarding our revised manuscript entitled "Relationships between distal sensorimotor polyneuropathy and cardiovascular complications in diabetic patients in the North-Eastern Hungarian region" (PONE-D-24-06839). Thank you for the opportunity to resubmit our revised manuscript to your journal.

Below you will find our point-by-point answers to the reviewer's questions. Changes to the revised manuscript were made in proofreading and marked in yellow. We also submit the final corrected version of the manuscript without markings.

Review Comments to the Author

Reviewer #1: The manuscript has improved after the correction, but I still have several comments to make.

Page 3 Line 62: You mean reduce the complications caused by diabetes or diabetic neuropathy? Please clarify.

Response: The Reviewer is absolutely right, we have corrected the sentence: „Early recognition of DSPN, modification of risk factors, appropriate glycaemic control, blood pressure control, maintenance of body weight, regular physical activity, treatment of hypertension and abnormal blood lipid levels can reduce the development of complications caused by diabetes mellitus.” (line 98)

Page 3 Line 70-74: Inappropriately elaborated here. This should be in your limitation to justify since you did not perform the tests.

Response: We added the following sentence to the "Limitation" section: „Sixth, as part of the diagnosis of neuropathy, we did not perform an electrophysiological examination (electroneurography) or determine the intraepidermal nerve fiber density from a skin biopsy.” (line 404-405)

The authors do not seem to understand the comments from multiple reviewers when we questioned how did they define their cohort of diabetic neuropathy. Page 5 Line 123-125 is redundant.

Response: In accordance with the Reviewer's request, the unnecessary sentence was deleted, and our group of patients with diabetic neuropathy was reworded as follows: „In our study, we selected diabetic patients in whom the typical symptoms of DSPN were observed, typically numbness, tingling, burning pain, sensitivity to touch, muscle weakness, and coordination disorders.” (line 160-162)

Page 5 Line 125-128: Grammarly incorrect. Please rephrase the whole sentence.

Response: Agreeing with the Reviewer's opinion, we rephrased the sentence: „A detailed evaluation of DSPN was performed in all participants using the DN4 (Douleur Neuropathique 4 Questions) and Neuropathy Symptom Score questionnaires during the screening of neuropathic pain.” (line 162-163)

Page 5 Line 133-134: This is redundant since you have justified using other tests plus Neurometer to define the neuropathy.

Response: In accordance with the Reviewer's request, the unnecessary sentence was deleted, and we rephrased the sentence: „The superficial sensation was evaluated with a monofilament test, the vibration detection with a tuning fork test, and the examination was completed by an instrumental evaluation of the temperature detection thresholds.” (line 165-167)

Significant discrepancies in the values presented in text (including abstract) and tables. Please align all the values, including decimal points.

Response: Thank you very much for the Reviewer's comment, we did not follow the multiple changes in the tables during the reviewer process exactly in the text, we modified these discrepancies in the "Abstract" and the "Results" (line 27-28, line 35-38, line 192, line 212-214, line 216, line 243-246)

Page 8 Line 177 and Abstract: HbA1c above 7% - Is this variable continuous or categorical when included in the model? If this is categorical, this has not been clearly defined in Methods.

Response: The parameter HbA1c above 7% is included as a continuous variable in the analysis, as is the parameter albumin/creatinine ratio above 3 mmol/mol. The HbA1c cut-off value set at 7% is arbitrary but adapted to the health insurance conditions in Hungary. This is because the Hungarian health funder considers a carbohydrate intake above this value to be insufficient for diabetics and allows for the funded prescription of modern antidiabetic medications. In contrast, international guidelines consider an HbA1c value between 6-8% acceptable, taking into account individual preferences, age, comorbidities, duration of diabetes, and other factors.

How did the authors decide which variables to be included in the multivariate logistic regression model? This is not clear from the statistical analysis. And, could that be the reason why they did not find the duration of diabetes to be significant? The significance of independent predictors is very much depended on the variables that you include in the model.

Response: In a multiple logistic regression model, the selected variables included parameters for carbohydrate homeostasis (HbA1c, duration of diabetes), microvascular complications (eGFR, albumin/creatinine ratio, diabetic retinopathy), and lifestyle and macrovascular complications (composite cardiovascular endpoint, high blood pressure, smoking). We have added the mentioned sentence in "Statistical analysis”. (line 183-187) We tested several multiple logistic regression models to examine the influence of diabetes duration, but in none of them did diabetes duration prove to be a significant predictor of DSPN.

Page 9 Paragraph 3: In the end if the authors still did not find the duration of diabetes to be significant in multivariate analysis, please explain. This was not explained in the discussion.

Response: We hypothesize that, despite the statistical difference, both patient groups can be considered long-standing diabetics, which may explain why the duration of diabetes did not prove to be a significant predictor of DSPN in the multivariable logistic regression model. Accordingly, we have added the mentioned sentence in "Discussion". (line 310-312)

Page 9 Paragraph 4: But your cohort significantly contains more T2DM than T1DM, how do you explain this?

Response: The statements in the "Discussion" section refer to the association between poor glycemic control and the risk of diabetic neuropathy in both type 1 and type 2 diabetes mellitus. However, effective treatment of diabetes mainly reduces the progression of DSPN in patients with T1DM, and for those with T2DM, glycemic control has only limited benefits. The results of our study are consistent with previous findings that hyperglycemia may be significantly associated with an increased risk of diabetic neuropathy. Accordingly, we have modified the mentioned section in "Discussion". (line 317-320)

Page 11 Line 266-267: How would it be your study implications when you did not actually perform the tests?

Response: The Reviewer is absolutely right, we have corrected the sentence: „Consenquently, the use of diabetic neuropathy screening tests may reveal peripheral nerve dysfunction in recently diagnosed diabetic patients, enabling more appropriate glycemic control and increased monitoring of other microvascular complications during diabetes care.” (line 378-383)

Thank you again for your thorough review of our manuscript and your valuable feedback!

Reviewer #2: (No Response)

Thank you again for reviewing our manuscript and supporting its acceptance!

---

## [Decision Letter · Decision Letter 2]

18 Jun 2024

Correlations between distal sensorimotor polyneuropathy and cardiovascular complications in diabetic patients in the North-Eastern region of Hungary

PONE-D-24-06839R2

Dear Dr. Sztanek,

We’re pleased to inform you that your manuscript has been judged scientifically suitable for publication and will be formally accepted for publication once it meets all outstanding technical requirements.

Kind regards,

Marcelo Arruda Nakazone, M.D., Ph.D.

Academic Editor

PLOS ONE

Additional Editor Comments (optional):

Reviewers' comments:

Reviewer's Responses to Questions

**Comments to the Author**

1. If the authors have adequately addressed your comments raised in a previous round of review and you feel that this manuscript is now acceptable for publication, you may indicate that here to bypass the “Comments to the Author” section, enter your conflict of interest statement in the “Confidential to Editor” section, and submit your "Accept" recommendation.

Reviewer #1: (No Response)

2. Is the manuscript technically sound, and do the data support the conclusions?

Reviewer #1: (No Response)

3. Has the statistical analysis been performed appropriately and rigorously? 

Reviewer #1: (No Response)

4. Have the authors made all data underlying the findings in their manuscript fully available?

Reviewer #1: (No Response)

5. Is the manuscript presented in an intelligible fashion and written in standard English?

Reviewer #1: (No Response)

6. Review Comments to the Author

Reviewer #1: Remove page 3 line 70-74

Remove page 5 line 131-133

Since the authors have clarified that the HbA1c is a continuous variable, then “higher HbA1c” would be more appropriate to replace “HbA1c above 7%” in Abstract Line 35 and Results Line 178.

7. PLOS authors have the option to publish the peer review history of their article (what does this mean?). If published, this will include your full peer review and any attached files.

Reviewer #1: No

---

## [Editor Report · Acceptance letter]

24 Jun 2024

PONE-D-24-06839R2 

PLOS ONE

Dear Dr. Ferenc, 

I'm pleased to inform you that your manuscript has been deemed suitable for publication in PLOS ONE. Congratulations! Your manuscript is now being handed over to our production team.

Kind regards, 

on behalf of

Professor Marcelo Arruda Nakazone 

Academic Editor

PLOS ONE